# Prophage-encoded *Hm-oscar* gene recapitulates *Wolbachia*-induced male-killing in the tea tortrix moth *Homona magnanima*

**Hiroshi Arai**[1,2]*, **Susumu Katsuma**[3], **Noriko Matsuda-Imai**[3], **Shiou-Ruei Lin**[4†], **Maki N Inoue**[2], **Daisuke Kageyama**[1]*

[1]National Agriculture and Food Research Organization, Tsukuba, Japan; [2]United Graduate School of Agricultural Science, Tokyo University of Agriculture and Technology, Fuchu, Japan; [3]Department of Agricultural and Environmental Biology, Graduate School of Agricultural and Life Sciences, The University of Tokyo, Tokyo, Japan; [4]Crop Environment Section, Tea and Beverage Research Station, Ministry of Agriculture, Taoyuan, Taiwan

**\*For correspondence:**
dazai39papilio@gmail.com (HA);
kageyama.daisuke423@naro.go.jp (DK)

**Present address:** †Pesticide Application Division, Agricultural Chemicals Research Institute, Ministry of Agriculture, Taichung City, Taiwan

**Competing interest:** The authors declare that no competing interests exist.

## eLife Assessment

Hardly anything is known about the genetic basis and mechanism of male-killing. Recently, a gene called oscar, in the bacterium *Wolbachia*, was implicated in killing male corn borer moths by interfering with moth genes that control sex determination and proper dosage of sex-specific genes. In this paper, the authors show that a distantly related oscar gene in another strain of *Wolbachia* kills male tea tortrix moths in a similar mechanism. This **valuable** study cements our understanding of the sophisticated way that *Wolbachia* kills male moths and butterflies (Lepidoptera) so early in their development. The conclusions are supported by **solid** evidence.

**Abstract** *Wolbachia* are maternally transmitted bacterial symbionts that are ubiquitous among arthropods. They can hijack host reproduction in various ways, including male-killing (MK), where the sons of infected mothers are killed during development. The recent discovery of MK-associated *Wolbachia* genes, i.e., oscar in *Ostrinia* moths and wmk in *Drosophila* flies, stimulates our interest in the diversity and commonality of MK mechanisms, which remain largely unclear. We recently discovered that a *Wolbachia* symbiont of the moth *Homona magnanima* carries an MK-associated prophage region encoding homologs of oscar (*Hm-oscar*) and wmk (wmk-1–4). Here, we investigated the effects of these genes in the native host. Upon transient overexpression, *Hm-oscar*, but not wmk, induced male lethality in *H. magnanima*, in contrast to our observations in *Drosophila*, where the wmk homologs, but not *Hm-oscar*, killed the males. *Hm-oscar* disrupted sex determination in male embryos by inducing a female-type *doublesex* splicing and impaired dosage compensation, recapitulating the *Wolbachia* phenotype. Cell-based transfection assays confirmed that *Hm-oscar* suppressed the function of *masculinizer*, the primary male sex determinant involved in lepidopteran dosage compensation. Our study highlights the conserved roles of oscar homologs in *Wolbachia*-induced lepidopteran MK and argues that *Wolbachia* have evolved multiple MK mechanisms in insects.

**eLife digest** Bacteria, viruses and other microbes are often thought of as external threats – but some also live inside animals, where they can be passed from mother to offspring. These types of microbes are commonly carried by insects and have various strategies for manipulating reproduction so they can spread through a population more easily.

One striking example is male killing, in which microbes selectively kill the male offspring of infected mothers during development. This results in a predominately female population, which can continue transmitting the microbe to future generations. Here, Arai et al. investigate whether *Wolbachia* – a group of bacteria that infect more than half of all insect species – employ a universal male-killing strategy, or whether they have evolved different mechanisms tailored to specific hosts.

The team focused on a region of the *Wolbachia* genome that encodes multiple candidate genes involved in male killing. First, Arai et al. explored whether a *Wolbachia* gene called *Hm-oscar* is responsible for male killing in the moth *Homona magnanima*. They found that over activating *Hm-oscar* in embryos not infected with *Wolbachia* increased male mortality, reproducing the effects of male killing, leading to a female-biased sex ratio. Further experiments confirmed that *Hm-oscar* exerts this effect by inhibiting *masculinizer*, a key gene necessary for male development.

Next, Arai et al. tested another gene called *wmk*, which *Wolbachia* use to kill the male offspring of fruit flies. However, manipulating this gene in *H. magnanima* had no impact on male survival, suggesting that *Wolbachia* use different male-killing genes in different insect species.

Understanding how *Wolbachia* affect insect reproduction sheds light on how bacteria interact with different host species. This knowledge also has practical applications. For instance, male-killing strategies could be leveraged for pest control to reduce populations of harmful insects without using pesticides. Future research may uncover additional *Wolbachia* genes involved in reproductive manipulation and explore their potential use in managing insect populations.

## Introduction

Arthropods commonly carry microbial symbionts that are passed from mother to offspring (*Hurst and Frost, 2015*; *Werren et al., 2008*; *Hurst, 2017*). The maternally transmitted bacteria, belonging to the genus *Wolbachia* (Alphaproteobacteria), are estimated to be present in at least 40% of all insect species, making them one of the most widespread endosymbiont genera on the planet (*Werren et al., 2008*; *Zug and Hammerstein, 2012*). *Wolbachia* have achieved evolutionary success by manipulating host reproduction through various means that enhance endosymbiont transmission (*Werren et al., 2008*). Such manipulation of the host's reproduction includes cytoplasmic incompatibility (CI), parthenogenesis, feminization, and MK, each of which affects the biological features, distribution, and evolution of the host. Among these strategies, MK directly skews the sex ratio of the host population toward females by killing male offspring of infected mothers during development. The lack of symbiont transmission through male hosts often leads to the evolution of MK. *Wolbachia* have been shown to induce MK in a wide range of insect taxa (e.g. dipterans, lepidopterans, and coleopterans) and other arthropods (*Hurst et al., 1999*; *Kageyama and Traut, 2004*; *Hurst et al., 2000*). Furthermore, various bacteria, viruses, and microsporidia induce MK phenotypes (*Kageyama et al., 2012*; *Fujita et al., 2020*; *Kageyama et al., 2023*; *Nagamine et al., 2023*), and recent studies have postulated that these microbes have evolved their MK ability independently (*Harumoto and Lemaitre, 2018*; *Kageyama et al., 2023*; *Nagamine et al., 2023*; *Arai et al., 2023b*).

The evolution and molecular mechanisms underlying *Wolbachia*-induced MK, observed across diverse insect taxa, have attracted considerable attention for decades. MK mechanisms are hypothesized to be linked to sex determination cascades regulating male and female differentiation in insects (*Hornett et al., 2022*). Indeed, some MK-inducing *Wolbachia* strains disrupt the sex-determination system in host moths (*Sugimoto and Ishikawa, 2012*; *Sugimoto et al., 2015*; *Fukui et al., 2015*; *Arai et al., 2023b*). Lepidopteran insects (moths and butterflies) generally have a female heterogametic sex chromosome system (e.g. WZ in females, ZZ in males) and employ dosage compensation, which equalizes the Z-chromosome-linked gene dose between sexes (*Kiuchi et al., 2014*). Dosage compensation is regulated by the *masculinizer* gene (*masc*), which is critical for male development. The lepidopteran sex determination system consists of multiple transcriptional regulators, some of

which exhibit sex-linked expression and/or splicing isoforms. The primary male sex determinant, *masc,* also regulates the downstream master sex determination gene, *doublesex* (*dsx*), which exhibits sex-dependent splicing isoforms (*dsxF* in females and *dsxM* in males) (*Kiuchi et al., 2014*). In *Ostrinia* and *Homona* male moths, MK-inducing *Wolbachia* strains impair dosage compensation by disrupting the expression of *masc*. Furthermore, they disrupt sex determination in male moths by inducing the 'female' isoform of *dsx* (*dsxF*), leading to a 'mismatch' between genetic sex (male: ZZ sex chromosome constitution) and phenotypic sex (female: based on *dsxF*), ultimately resulting in male death (*Arai et al., 2023b*; *Fukui et al., 2015*; *Sugimoto et al., 2015*; *Sugimoto and Ishikawa, 2012*).

More recently, a *Wolbachia* protein named Oscar (Osugoroshi protein containing CifB C-terminus-like domain and many Ankyrin Repeats; Osugoroshi translates to male-killing in Japanese) was shown to recapitulate the *Wolbachia* wFur-induced MK phenotype in the native host moth *Ostrinia furnacalis* (*Katsuma et al., 2022*). The Oscar protein interacts with and degrades Masc protein, leading to the failure of dosage compensation and the production of female-type *dsx* isoforms in *Ostrinia* male moths (*Fukui et al., 2024*; *Katsuma et al., 2022*). Although *oscar* homologs are common in MK *Wolbachia* strains from Lepidoptera, MK strains from Diptera (*Drosophila*) and from several lepidopteran hosts lacked the gene (*Arai et al., 2023a*; *Arai et al., 2024a*). Furthermore, *oscar* homologs are evolutionarily dynamic, with highly variable sequences and structures (*Arai et al., 2024a*), making it difficult to assess their functional relevance. In addition to *oscar*, the helix-turn-helix domain-containing putative transcriptional regulator *wmk* is widely conserved among *Wolbachia* strains and induces various toxicities in *Drosophila* flies, ranging from no effect, weak male lethality (30% mortality), strong male lethality (90–100% mortality), or complete lethality of both sexes (*Perlmutter et al., 2019*; *Perlmutter et al., 2020*; *Perlmutter et al., 2021*; *Arai et al., 2023a*). Although the mechanisms underlying the *wmk*-induced toxicities and their connection to sex determination systems remain unclear, these findings suggest that *Wolbachia* strains carry multiple factors that cause male lethality. However, the diversity and commonality of these functions in insects remain largely unknown, partly due to the technical challenges in validating gene functions in non-model insects.

We recently discovered a prophage region responsible for a *Wolbachia*-induced MK by comparing closely related non-MK *Wolbachia* (*w*Hm-c) and MK *Wolbachia* (*w*Hm-t) in the tea tortrix moth *Homona magnanima* (Tortricidae) (*Arai et al., 2023a*; *Arai et al., 2024b*). The MK-associated prophage element encodes potential MK causative genes–four *wmk* homologs (*wmk*-1 to *wmk*-4) as well as an *oscar* homolog (*Hm-oscar*, encoding 1181 aa protein), which differs significantly in sequence length and structure from the *w*Fur-encoded *oscar* (encoding 1830 aa protein) (*Figure 1a*, *Figure 1—source data 1*). When these MK candidate genes are transgenically overexpressed in *Drosophila*, *wmk-1* and *wmk-3* have a lethality of almost 100%, while *wmk-2*, *wmk-4*, and *Hm-oscar* induce no lethal effects when singly overexpressed. Furthermore, co-expression of the adjacent *wmk-3* and *wmk-4* has been shown to induce the death of 90% of male flies and restores female survival, suggesting their combined action (*Arai et al., 2023a*). However, transgenic expression of genes from a moth-derived male killer in flies does not necessarily replicate the native host–bacteria interaction, and the mechanistic links between prophage-encoded *Wolbachia* genes and MK in the native host *Homona* remain unclear.

In this study, we showed that the prophage-encoded *Hm-oscar* recapitulates *Wolbachia*-induced MK in *H. magnanima*. Furthermore, we clarified the mechanistic links to host sex-determination cascades both in vivo and in vitro and discussed the underlying mechanisms of MK in Lepidoptera, arguing for the diverse evolutionary origin of *Wolbachia*-induced MK.

## Results
### *Hm-oscar* induces female-biased sex ratios

To achieve the transient overexpression of *Hm-oscar* and the four *wmk* genes (*wmk*-1, *wmk*-2, *wmk*-3, and *wmk*-4), constructed mRNA (cRNA) was injected into *Wolbachia*-free *H. magnanima* embryos. Subsequently, the adult moths that emerged from the cRNA-injected embryos were sexed based on their external morphology. When *Hm-oscar* was overexpressed, the sex ratio of adults was strongly female-biased (42 males and 218 females in total, 14.4 ± 11.1% male ratio in 15 replicates), which was in sharp contrast with the ratios observed in the *GFP*-injected (126 females and 123 males in total, 49.1 ± 2.08% male ratio in 10 replicates) and non-injected (NSR) (201 females and 215 males in total,

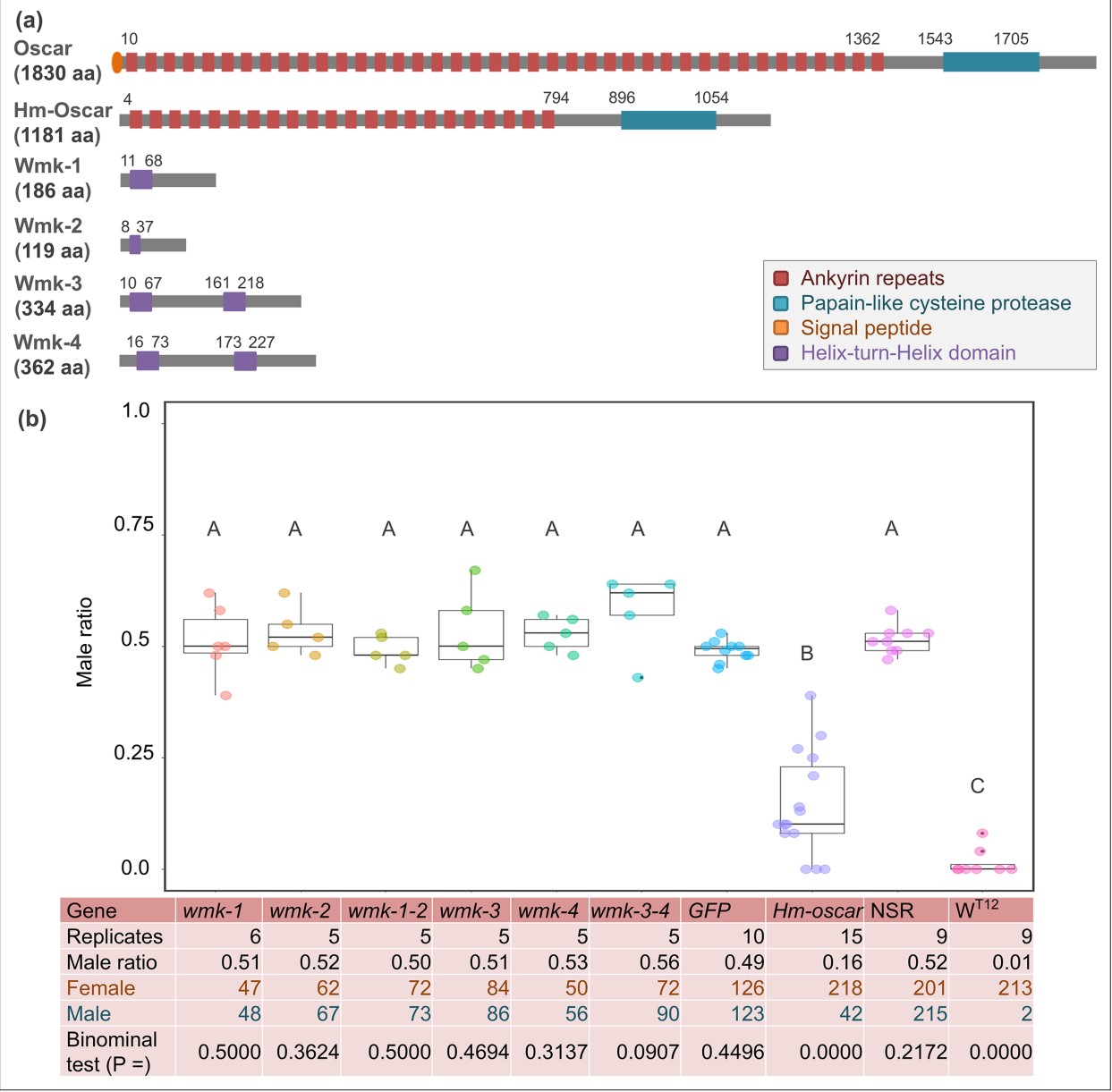

**Figure 1.** Transient expressions of the *homologs of oscar (Hm-oscar)* gene resulted in female-biased sex ratios. (**a**) Structures of *w*Fur-encoded Oscar and *w*Hm-t-encoded Hm-Oscar and four Wmk proteins. (**b**) Male ratio of adult progeny obtained from cRNA-injected groups (*wmk-1*, *wmk-2*, T2A-bridged *wmk-1* and *wmk-2*, *wmk-3*, *wmk-4*, T2A-bridged *wmk-3* and *wmk-4*, GFP, and *Hm-oscar*; n = 5–15 independent replicates using different egg masses), the non-injected normal sex ratio (NSR) line, and the male-killing (MK) *w*Hm-t-positive W^T12 line. The total numbers of adult females and males are shown at the bottom. Different letters indicate significant differences (Steel–Dwass test, p<0.05). The dot plots show all data points individually.

The online version of this article includes the following source data for figure 1:

**Source data 1.** Sequence ID and data for *wmk* and *oscar* homologs displayed in *Figure 1a*.

**Source data 2.** Number of males and females and sex ratio displayed in *Figure 1b*.

**Source data 3.** Statistical analysis (Steel-Dwass test) data displayed in *Figure 1b*.

51.4 ± 3.08% male ratio in nine replicates) groups (p=0.001 and 0.002, respectively, Steel–Dwass test, *Figure 1b*, *Figure 1—source data 2*, *Figure 1—source data 3*). Compared with the *GFP*-injected group, the sex ratio was not biased by the overexpression of *wmk*-1 (51.1 ± 7.27% male ratio in six replicates, p=0.999), *wmk*-2 (53.2 ± 4.77% male ratio in 6 replicates, p=0.810), *wmk*-3 (53.3 ± 7.91% male ratio in five replicates, p=0.999), and *wmk*-4 (52.9 ± 3.47% male ratio in five replicates, p=0.773). Although the dual expression of *wmk*-3 and *wmk*-4 induces strong male lethality in *Drosophila* (***Arai***

*et al., 2023a*), here, it did not skew the sex ratio of *Homona* compared with the ratio detected in the *GFP*-injected group (57.9 ± 7.67% male ratio in five replicates, p=0.704). Similarly, the dual expression of the tandemly arrayed *wmk*-1 and *wmk*-2 did not bias the sex ratio (49.4 ± 2.82% male ratio in five replicates) compared with the value in the *GFP*-injected group (p=1.000). The sex ratio data of all replicates are available on the *Figure 1—source data 2*.

## Males are killed mainly at the embryonic stage

The sex of hatched larvae (neonates) and the remaining unhatched embryos was determined by the presence or absence of W chromatin, a condensed structure of the female-specific W chromosome observed during interphase. MK *Wolbachia w*Hm-t kills *Homona* males during embryogenesis, resulting in a female-biased sex ratio in hatched larvae (neonates) and a male-biased sex ratio in unhatched embryos (*Arai et al., 2023a*; *Arai et al., 2020*). In line with this finding, the sex ratios of hatched larvae (34 females and 6 males) and unhatched embryos (27 females and 47 males) of the MK *w*Hm-t infected line (W$^{T12}$) were also female- and male-biased, respectively (p=0.013 and 0, respectively). In the *Hm-oscar*-injected group, the sex ratio of the hatched larvae (neonates) was strongly female-biased (21 females and 3 males, p=0.0001 in the binomial test) (*Figure 2a*, *Figure 2— source data 1*). In contrast, the unhatched mature embryos showed a male-biased ratio (18 females and 38 males, p=0.005), suggesting that male mortality primarily occurred during the embryonic stage, reflecting the *w*Hm-t-induced MK phenotype. In the non-injected group (NSR), the sex ratios of hatched larvae (28 females and 28 males) and unhatched embryos (23 females and 25 males) were not biased (p=0.553 and 0.443, respectively).

## Female-type sex determination in male embryos that are destined to die

Splicing patterns of the downstream sex determinant *dsx* were assessed in each of the *H. magnanima* embryos that were *Hm-oscar*-expressed, *GFP*-expressed, *w*Hm-t-infected, or non-expressed (i.e. non-injected, NSR). In *GFP*-expressed and NSR embryos, females and males (which were identified by the presence and absence of W chromatin, respectively) exhibited a female- and male-type splicing variant of *dsx*, respectively. However, both *w*Hm-t-infected and *Hm-oscar*-expressed embryos induced female-type *dsx* splicing regardless of the presence or absence of W chromatin (*Figure 2b*, *Figure 2—source data 2*, *Figure 2—source data 3*). Notably, male embryos expressing *Hm-oscar* also exhibited weak male-type *dsx* splicing in addition to the female-type splicing, resembling the previously observed pattern in male embryos infected with low-titer *w*Hm-t (*Arai et al., 2023b*).

## *Hm-oscar* impairs dosage compensation in male embryos

To confirm the effects of *Hm-oscar* on dosage compensation, we quantified the expression of Z-linked genes using RNA-seq and qPCR. Unlike other lepidopteran species, the family Tortricidae, including *H. magnanima*, generally possess a large Z chromosome that is homologous to *B. mori* chromosomes 1 (Z) and 15 (autosome). RNA-seq analysis revealed that, in *Hm-oscar*-injected embryos, Z-linked genes (homologs on the *B. mori* chromosomes 1 and 15) were more expressed in males than in females (*Figure 3a*, *Figure 3—source data 1*, *Figure 3—source data 2*), which was not observed in the *GFP*-injected group (*Figure 3b*, *Figure 3—source data 3*, *Figure 3—source data 4*). Similarly, as previously reported by *Arai et al., 2023b*, high levels of Z-linked gene expression were also observed in *w*Hm-t-infected males, but not in NSR males (*Figure 3c and d*, *Figure 3—source data 5*, *Figure 3—source data 6*, *Figure 3—source data 7*, *Figure 3—source data 8*). The high (i.e. doubled) Z-linked gene expression in both *Hm-oscar*-expressed and *w*Hm-t-infected males was further confirmed by quantification of the Z-linked *Hmtpi* gene (*Figure 3c*, *Figure 3—source data 9*, *Figure 3—source data 10*). These trends were statistically supported, with all data available in the source data files (i.e. *Figure 3*, *Figure 3—source data 1*, *Figure 3—source data 2*, *Figure 3—source data 3*, *Figure 3— source data 4*, *Figure 3—source data 5*, *Figure 3—source data 6*, *Figure 3—source data 7*, *Figure 3—source data 8*, *Figure 3—source data 9*, and *Figure 3—source data 10*).

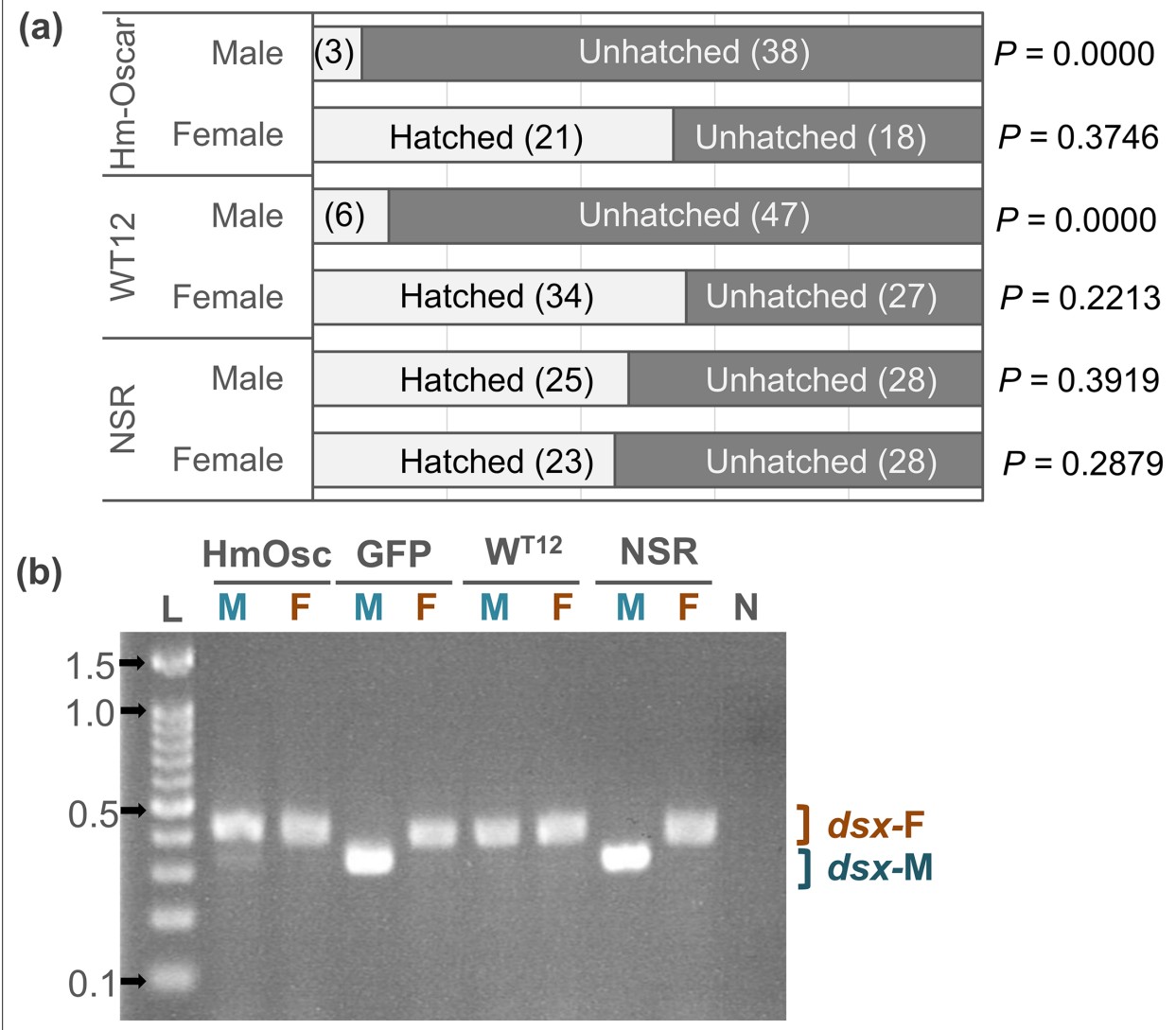

**Figure 2.** Homologs of oscar (Hm-Oscar) induced lethality in male embryos with the female-type sex determination. (**a**) Sex ratio of the hatched larvae and unhatched embryos in the Hm-oscar-expressed, wHm-t-infected, and non-infected/expressed groups. Females and males were discriminated based on the presence or absence of W chromatin. The number of individuals is indicated in brackets. (**b**) Splicing patterns of the downstream sex-determining gene dsx of H. magnanima embryos (5 d post oviposition). Abbreviations: HmOsc, Hm-oscar injected group; GFP, GFP-injected group; W^T12, wHm-t-infected line; NSR, non-infected/injected line. M and F indicate W chromatin-negative (ZZ: male genotype) and W chromatin-positive (ZW: female genotype) mature embryos, respectively. dsx-F and dsx-M represent female and male-specific splicing variants, respectively. L: 100 bp DNA ladder (ExcelBand 100 bp DNA Ladder, SMOBIO Technology, Inc, Hsinchu, Taiwan). 0.1, 0.5, 1.0, and 1.5 kb markers are indicated with arrows. N: negative control (water).

The online version of this article includes the following source data for figure 2:

**Source data 1.** Number of males and females displayed in **Figure 2a**.

**Source data 2.** Original gel image for band patterns of dsx displayed in **Figure 2b**.

**Source data 3.** PDF file containing an original gel image for **Figure 2b**, indicating the relevant bands and treatments.

### Hm-oscar suppresses the masculinizing functions of lepidopteran masc genes

To confirm whether Hm-oscar suppresses the functions of the primary male sex determinant masc, their interactions were assessed through transfection in BmN-4 cells. These cells, derived from B. mori ovaries (**Grace, 1967**), exhibit female-type default sex determination (**Kiuchi et al., 2014**). However, overexpression of the masc gene from various lepidopteran insects, such as O. furnacalis,

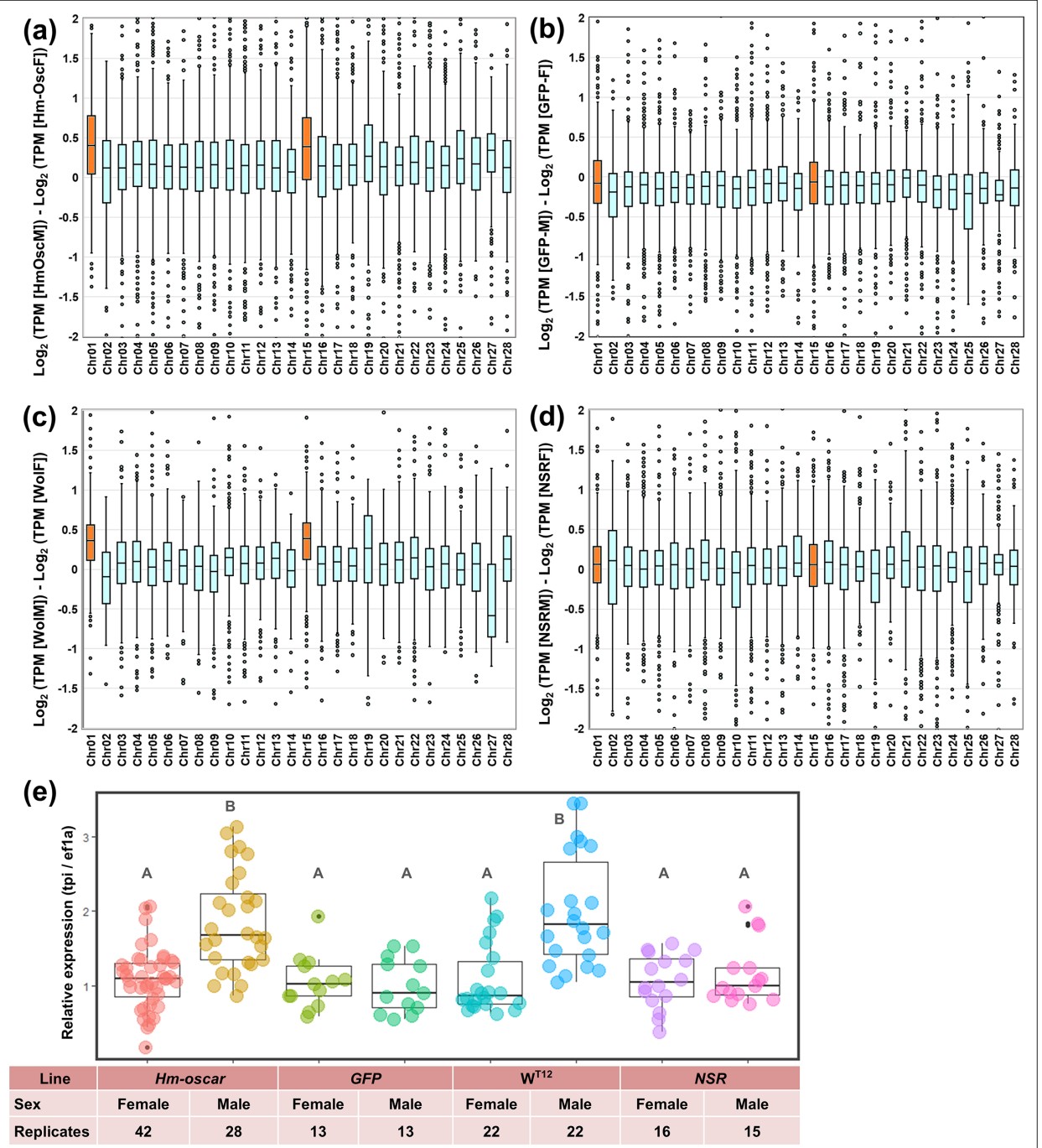

**Figure 3.** *Homologs of oscar (Hm-oscar)*-overexpressed male embryos showed higher levels of Z-linked gene expression. (**a–d**) Normalized expression levels (TPM) and chromosomal distributions of transcripts in *H. magnanima* embryos. RNA-seq data of embryos (108 hpo) were used to make the following comparisons: *Hm-oscar*-injected males versus *Hm-oscar* injected females (**a**), GFP-injected males versus GFP-injected females (**b**), W$^{T12}$ males versus W$^{T12}$ females (**c**), and normal sex ratio (NSR) males versus NSR females (**d**). The chromosome number for each *H. magnanima* transcript-derived contig was assigned based on *Bombyx mori* gene models. The x-axis represents the chromosome number of *B. mori* (shown as chr01 to chr28), and chr01 and chr15 (highlighted in orange) correspond to the Z chromosome of *H. magnanima* (**Arai et al., 2023b**). (**e**) Relative expression of a Z-linked *tpi* gene in *H. magnanima* groups. Statistical significance is highlighted with different alphabets.

The online version of this article includes the following source data for figure 3:

**Source data 1.** Expression data displayed in *Figure 3a*.

**Source data 2.** Statistical analysis (Steel-Dwass test) data displayed in *Figure 3a*.

*Figure 3 continued on next page*

*Figure 3 continued*

**Source data 3.** Expression data displayed in *Figure 3b*.

**Source data 4.** Statistical analysis (Steel-Dwass test) data displayed in *Figure 3b*.

**Source data 5.** Expression data displayed in *Figure 3c*.

**Source data 6.** Statistical analysis (Steel-Dwass test) data displayed in *Figure 3c*.

**Source data 7.** Expression data displayed in *Figure 3d*.

**Source data 8.** Statistical analysis (Steel-Dwass test) data displayed in *Figure 3d*.

**Source data 9.** Expression data displayed in *Figure 3e*.

**Source data 10.** Statistical analysis (Steel-Dwass test) data displayed in *Figure 3e*.

induces male-type sex determination by promoting the expression of $Bmdsx^M$ and $BmIMP^M$ in BmN-4 cells (*Fukui et al., 2015*; *Katsuma et al., 2019*; *Katsuma et al., 2022*). Since $BmIMP^M$ regulates the splicing of $Bmdsx^M$ and is exclusively expressed in males, we selected $BmIMP^M$ as a quantitative marker of masculinization, as described in *Katsuma et al., 2022*.

In contrast with the non-expressed control groups, which exhibited default female-type sex determination, cells expressing *Hmmasc* exhibited male-type sex determination (*Figure 4a*, *Figure 4—source data 1*). This was indicated by the increased expression of the male-specific splicing variant $BmIMP^M$, although this increase was not statistically significant. Furthermore, the masculinizing function of *Hmmasc* was suppressed when *Hmmasc* and *Hm-oscar* were co-expressed, as evidenced by

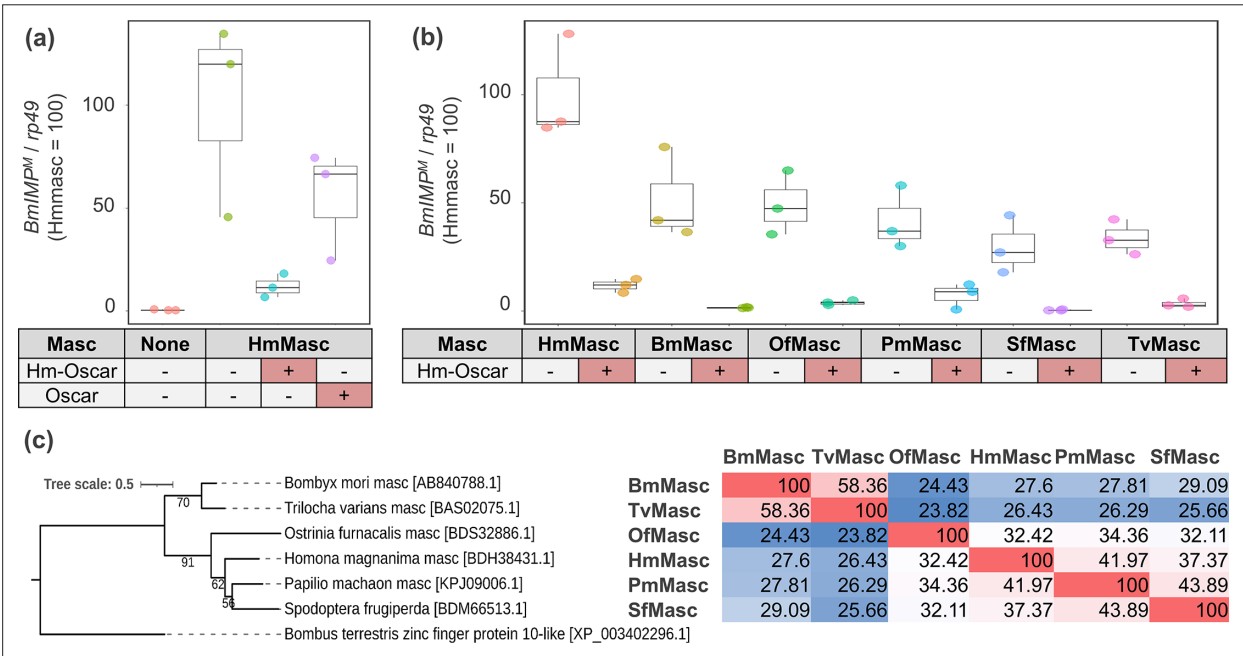

**Figure 4.** Hm-oscar suppressed the masculinizing function of lepidopteran Masc. (**a**) Relative expression levels of the male-specific $BmImp^M$ variant in transfected BmN-4 cells. The relative expression of $BmImp^M$ in the control (non-inserted pIZ/V5) and *Hm-oscar*/*oscar* co-transfected groups was normalized by setting the mean in the *Hmmasc* single transfected group as 100. (**b**) Relative expression levels of the $BmImp^M$ in *Hm-oscar/masc* transfected BmN-4 cells. The relative expression of $BmImp^M$ in each condition was normalized by setting the mean in the *Hmmasc* single transfected group as 100. The dot plots show all data points individually. Each experimental condition was replicated three times. Abbreviations: Hm, *Homona magnanima*; Bm, *Bombyx mori*; Of, *Ostrinia furnacalis*; Pm, *Papilio machaon*; Sf, *Spodoptera frugiperda*; and Tv, *Trilocha varians*. (**c**) Phylogeny and homologies of the Masc protein sequences used in this study. The sequence homology between each pair is presented in the figure and visualized as a heatmap, with red indicating high homology (Max = 100) and blue indicating low homology (Min = 23.82).

The online version of this article includes the following source data for figure 4:

**Source data 1.** Expression data for IMP gene displayed in *Figure 4a*.

**Source data 2.** Expression data for IMP gene displayed in *Figure 4b*.

**Source data 3.** Sequence homology data displayed in *Figure 4c*.

the low expression levels of *BmImp*^M (*Figure 4a*). *w*Fur-encoded Oscar, which differs from Hm-Oscar in structure and amino acid sequence (*Figure 1a*), also suppressed *Hmmasc*, but its function was less active than that of *Hm-oscar* (*Figure 4a*). *Hm-oscar* also suppressed the function of *masc* genes derived from various lepidopteran insects (i.e. *O. furnacalis*, *Spodoptera frugiperda*, *Bombyx mori*, *Trilocha varians*, and *Papilio machaon*), which are highly divergent in their amino acid sequences (*Figure 4b, c*, *Figure 4—source data 2*, *Figure 4—source data 3*). These results suggest that *Hm-oscar* has a broad spectrum of activity in Lepidoptera.

## Discussion

In this study, we showed that the phage-encoded *Hm-oscar*, but not *wmk*, induced male lethality and a female-biased sex ratio in *H. magnanima*. Furthermore, the overexpressed *Hm-oscar* impaired male sex determination in *Homona*, recapitulating the *w*Hm-t-induced phenotypes. Cell-based assays confirmed that *Hm-oscar* suppressed the masculinizing functions of *masc*. These results strongly suggested that *Hm-oscar* underlies the *w*Hm-t-induced MK function in *H. magnanima*. In light of our results and previous findings in *Homona*, *Ostrinia*, and *Bombyx* (*Arai et al., 2023a*; *Katsuma et al., 2022*; *Kiuchi et al., 2014*), we hypothesize the following molecular mechanisms for *w*Hm-t-induced MK: The Hm-Oscar protein targets and inhibits the function of the HmMasc protein, thereby resulting in female sex determination and disrupting dosage compensation, which ultimately leads to male death. Although the full mechanisms remain unclear, unbalanced Z-linked gene expression mediated by improper dosage compensation—resulting in Z-linked gene expression levels twice as high as those in normal males—appear to be lethal for lepidopteran males (*Kiuchi et al., 2014*; *Fukui et al., 2015*; *Visser et al., 2021*).

The *Wolbachia*-encoded *oscar* homologs identified so far show diversity in sequence length and are classified as type I, II, and III based on their amino acid sequence homologies. Type I and II Oscar proteins, derived from MK-inducing *Wolbachia*, contain many ankyrin repeats and a proteinase domain (*Arai et al., 2024a*). Although long ankyrin repeat sequences are postulated to be critical for the function of *w*Fur-encoded Oscar protein (1830 aa, type I) (*Katsuma et al., 2022*), our study revealed that the functions of Hm-Oscar protein (1181 aa, type II), which encodes fewer ankyrin repeats, were comparable with those of Oscar protein carried by *w*Fur. Interestingly, type II *oscar* gene is also present in the feminizing *Wolbachia w*Fem in *Eurema* butterflies (*Arai et al., 2024a*). *oscar* homologs, which inhibit the masculinizing function and induce female sex determination, may have a conserved function in *Wolbachia*-induced MK and feminization in Lepidoptera.

In contrast to the results of this study, we have previously demonstrated that the phage-encoded *wmk*, but not *Hm-oscar*, induces male lethality in *D. melanogaster* (*Arai et al., 2023b*). Although the means of expression are different (i.e. transient in *Homona* and transgenic in *Drosophila*), this finding highlighted the differences in the mode/range of action of *Wolbachia* genes between insect species. It has been hypothesized that microbes induce MK in insects by targeting molecular mechanisms involved in sex determination and differentiation (*Hornett et al., 2022*). Sex determination systems in insects are diverse; for example, Lepidoptera (including *H. magnanima*) and Diptera (including *D. melanogaster*) do not share any known sex-determining genes other than *dsx* (*Suzuki, 2018*). The different outcomes in *Homona* and *Drosophila* are probably due to their different sex determination systems. Because *oscar* interacts with and suppresses *masc*, *Hm-oscar* could induce mortality in *Homona* males that possess *masc*, but not in *Drosophila* males that lack it. Considering that *oscar* homologs are not present in known MK *Wolbachia* strains in dipteran insects (*Arai et al., 2024c*; *Katsuma et al., 2022*), the mechanisms (i.e. causative genes) of *Wolbachia*-induced MK probably differ between insect taxa (e.g. between Lepidoptera and Diptera). While the mechanisms underlying *wmk*-induced lethality remain unclear, the distinct effects associated with this gene between *Homona* and *Drosophila* may also reflect their genetic backgrounds, such as the presence or absence of host factor(s) that interact with *wmk*. In addition, *Katsuma et al., 2022* reported that the *wmk* homologs encoded by *w*Fur did not affect the masculinizing function of *masc* in vitro, indicating that *wmk* likely targets factors other than *masc*. Whilst we cannot rule out the possibility that *wmk* may work synergistically or interactively with *oscar* in vivo—potentially acting downstream of *oscar*'s impact—our results strongly suggested that *Wolbachia* strains have acquired multiple MK genes through evolution.

*Wolbachia*-induced phenotypes are known to be influenced by the genetic backgrounds of hosts (*Hornett et al., 2006*; *Sasaki et al., 2002*; *Veneti et al., 2012*). Our study showed that the

wHm-t-encoded *Hm-oscar* suppresses the function of *masc* in *H. magnanima* more efficiently than the *w*Fur-encoded *oscar*, suggesting that this *Wolbachia* factor has undergone evolutionary tuning to adapt to its natural host. However, the mere presence of *oscar* and *wmk* homologs does not ensure MK expression. For example, *w*CauA, bearing type I *oscar*, did not induce MK in *Ephestia* (*Cadra*) *cautella* collected around 2000, although it did induce MK when transferred to the closely related host *Ephestia kuehniella* (**Sasaki et al., 2002**). An MK phenotype, presumably induced by *w*CauA, was also recorded in *E. cautella* around the 1970s (**Takahashi and Kuwahara, 1970**). These findings suggest a suppressor evolution against *w*CauA-induced MK in *E. cautella*. In addition, *Wolbachia* typically encode a number of *wmk* homologs in their genomes, but their effects differ among insects and are not apparent in some species such as *Homona*, possibly due to the diversity in sex determination system or the evolution of suppressors. Nonetheless, given the high copy number of *wmk* homologs in *Wolbachia* genomes, they may still contribute to *Wolbachia* biology as transcriptional regulators and potential virulence factors. Virulence genes often undergo duplication and substitution under strong selective pressure (**Hill et al., 2022**; **Jones and Dangl, 2006**). An intense evolutionary arms race between *Wolbachia* and their hosts could have increased and diversified the MK-associated *Wolbachia* genes. Conversely, natural selection favors the rescue of males by suppressing the *Wolbachia*-induced reproductive manipulations (**Hornett et al., 2006**; **Hornett et al., 2014**), which may involve changes in the sex determination system because *Wolbachia* strains frequently hijack host reproduction systems. In this context, *Wolbachia* and other MK-inducing microbes may be a hidden driver for the diversification of complex insect sex determination systems.

In this study, we clarified the conserved roles of the *Wolbachia*-encoded *oscar* homologs in Lepidoptera and demonstrated that *Wolbachia* have evolved distinct MK mechanisms (through causative genes) in insect taxa. The diversification of phenotype/virulence-associated genes and the rampant horizontal transmission of phages carrying virulence genes between *Wolbachia* strains may have contributed to the outstanding success of this bacterial genus. In addition to *oscar* and *wmk*, *Wolbachia* may retain other uncharacterized genes that induce male lethality, and further studies on diverse *Wolbachia*–host systems are highly warranted. Our findings provide insights into the molecular mechanisms and evolutionary relationships between endosymbionts and their hosts, which may also contribute to the design of pest management strategies.

## Materials and methods
### Experimental model and subject details
#### *Homona magnanima*
A laboratory-maintained *H. magnanima* line with a normal sex ratio (NSR) that was negative for *Wolbachia* and other endosymbionts, was used in our experiments. This line was initially collected in Tokyo, Japan, in 1999 and has been maintained inbred for over 250 generations in the laboratory. Larvae were reared using the artificial SilkMate 2 S diet (Nosan Co., Yokohama, Japan) at 25 °C under a long photoperiod (16 L:8D) (**Arai et al., 2019**).

The laboratory-maintained all female *H. magnanima* W$^{T12}$ line, which was initially collected in Taiwan (Tea Research Extension Station, Taoyuan city) in 2015 (**Arai et al., 2020**), was also used in this study. This line was maintained for over 50 generations by crossing it with the males of the NSR line.

#### *BmN-4* cell line
*Bombyx mori* BmN-4 cells (provided by Chisa Yasunaga-Aoki, Kyushu University, and maintained in our laboratory) were cultured at 26 °C in IPL-41 Insect Medium (Applichem, Darmstadt, Germany) supplemented with 10% fetal bovine serum. The cells used were authenticated by sequencing *masc* gene. The BmN-4 cells are of insect origin; therefore, we did not test for mycoplasma contamination, which is commonly screened in mammalian cell cultures. However, to prevent potential contamination, we used cells that were passaged in a medium containing penicillin, streptomycin, and amphotericin B.

### Transient expression of MK-associated phage genes
#### mRNA synthesis
Codon-optimized *wmk* genes (*wmk-1* to *wmk-4*), conjugated *wmk* pairs using a T2A peptide (i.e. *wmk*-1-T2A-*wmk*-2 and *wmk*-3-T2A-*wmk*-4), and *Hm-oscar* genes synthesized by **Arai et al., 2023a** were

used for mRNA synthesis. These synthetic genes were ligated into the plasmid pIZ/V5-His (Invitrogen, MA, USA) using the NEBuilder HiFi DNA Assembly kit (New England Biolabs, MA, USA) following the manufacturer's protocol. The inserts of the vector (i.e. *wmk*-1, *wmk*-2, *wmk*-3, *wmk*-4, *Hm-oscar*, *wmk*-1-T2A-*wmk*-2, *wmk*-3-T2A-*wmk*-4, and *GFP* as a control gene) were amplified using KOD-one (TOYOBO, Osaka, Japan) with the primer set containing the T7 promoter described in *Fukui et al., 2015* (i.e. pIZ-F-T7: 5'-TAATACGACTCACTATAGGGAGACAGTTGAACAGCATCTGTTC-3' and pIZ-R: 5'-GACAATACAAACTAAGATTTAGTCAG-3') under the following PCR conditions: 20 cycles at 98 °C for 10 s, 62 °C for 5 s, and 68 °C for 15 s. The amplicons were purified using the QIAquick PCR purification kit (Qiagen, Hilden, Germany), and 100–200 ng DNA was used for mRNA synthesis. The capped mRNA (cRNA) with poly(A) tail was synthesized using the mMESSAGE mMACHINE T7 Ultra kit (Invitrogen, MA, USA) following the manufacturer's protocol with some modifications. In brief, the assembled transcription reaction (10 µL of T7 2 X NTP/ARCA, 2 µL of 10X T7 reaction buffer, 2 µL of T7 enzyme mix, and 6 µL of linear DNA template diluted with nuclease-free water) was incubated at 37 °C for at least 10 hr to maximize RNA yields. After poly(A) tailing, cRNA was purified using ISOGEN II (Nippongene, Tokyo, Japan) and dissolved in up to 10 µL of nuclease-free water to achieve an RNA concentration of approximately 1500–2000 ng/µL. The synthesized cRNA was preserved at −80 °C until further use.

## Preparation of *H. magnanima* eggs

A total of 15 males and 10 females of *H. magnanima* were mated in a plastic box (30 cm × 20 cm × 5 cm) for 3–4 d. The collection of egg masses began the day after oviposition was confirmed. In brief, newly oviposited egg masses were collected at 30 min intervals during the dark period using red light (which the moths cannot perceive). Females started to oviposit eggs at least 4 hr into the dark period and, within less than 30 min post oviposition (mpo), the egg masses were collected. The collection lasted until the start of the light period. The egg masses were then subjected to microinjection assays.

## Inoculum preparation, microinjection, and maintenance of the injected embryos

A glass needle for microinjection was prepared using glass capillary GD-1 (Narishige, Tokyo, Japan) with a PC-10 puller (Narishige). The glass capillaries were pulled at two temperatures (first stage: 75 °C, second stage: 65 °C) using two heavy weights and one light weight. The movement position during the second heating stage was set to 3 mm (range 1–10 mm).

The cRNA solution was diluted in a buffer (100 mM potassium acetate, 2 mM magnesium acetate, and 30 mM HEPES-KOH; at pH 7.4) containing 0.2% (W/V) Brilliant Blue FCF (Wako, Osaka, Japan) to obtain an RNA concentration of 1000 ng/µL. Approximately 1–4 µL of dye-containing mRNA solution was aspirated into the glass needles (capillaries), and the edge of each needle was ground using Micro Grinder EG-402 (Narishige) at an angle of 20 degrees for 2 s.

The fresh egg masses of *H. magnanima* (collected at less than 30 mpo as described above) were put on a double-sided sticky tape (15 mm × 5 m T4612, Nitoms, Tokyo, Japan) and placed on glass slides. Under a Nikon SMZ1270 microscope (Nicon, Tokyo, Japan), the RNA solution (30–100 fL) was injected into individual eggs using an microinjector IM-400 (Narishige) (balance pressure set at 0.010–0.030; injection pressure set at 0.050–0.100) and a QP-3JOY-2R electric micromanipulator (MicroSupport, Shizuoka, Shizuoka).

The injected egg masses were maintained in a 90 mm plastic Petri dish fitted with a slightly wet filter paper. As high humidity interfered with the development of the embryos, the injected eggs (egg mass) were first maintained at a relative humidity of 30–50% (0–3 d post-injection) and then at a higher humidity until hatching (4–6 dpo, 60% relative humidity). The hatched larvae were reared separately with 1/2 ounces of SilkMate 2 S (Nosan Co.) until eclosion. The adult *H. magnanima* moths that emerged from the cRNA-injected embryos were sexed based on their external morphology.

## Sexing of embryos/neonates and RNA extraction

To verify the effect of the transient expression of *Hm-oscar* on sex determination in *H. magnanima*, *Hm-oscar*/*GFP*-expressed, non-injected, and *w*Hm-t-infected mature embryos showing black head capsule (1 d before hatching, 5–6 d post oviposition, dpo) were dissected on glass slides using forceps, as described in *Arai et al., 2022*. Their Malpighian tubules were fixed with methanol/acetic acid

(50% v/v) and stained with lactic acetic orcein for W chromatin observations. The remaining tissues not used for sexing were stored in ISOGEN II (Nippon Gene) at −80 °C until subsequent extraction. In total, 12 males or females (confirmed based on the presence or absence of W chromatin) were pooled and homogenized in ISOGEN II to extract RNA as described in *Arai et al., 2023b*. In brief, 0.4 times the volume of UltraPure distilled water (Invitrogen) was added to the ISOGEN II homogenates, which were centrifuged at 12000×*g* for 15 min at 4 °C to pellet proteins and DNAs. The resulting supernatant was mixed with the same volume of isopropanol to precipitate RNAs; then, the resulting solutions were transferred to EconoSpin columns (Epoch Life Science) and centrifuged at 17,900×*g* and 4 °C for 2 min. The RNAs captured in the column were washed twice with 80% ethanol and eluted in 20 μL of UltraPure distilled water (Invitrogen).

### *Hmdsx* detection

Sex-specific *dsx* splicing variants of *H. magnanima* were detected as described in *Arai et al., 2023b*. In brief, total RNA (100–300 ng) extracted from sex-determined mature embryos was reverse-transcribed using PrimeScript II Reverse Transcriptase (TaKaRa, Shiga, Japan) at 30 °C for 10 min, 45 °C for 60 min, and 70 °C for 15 min. Then, cDNA was amplified using KOD-FX Neo (Toyobo Co., Ltd.) with the following two primers: Hmdsx_long3F (5′-TGCCTAAAGTGAAAACGCCGAGGAGCC-3′) and Hmdsx_Mrev (5′-TGGAGGTCTCTTTTCATCCGG-3′). The PCR conditions used were as follows: 94 °C for 2 min, followed by 45 cycles of 98 °C for 10 s, 66 °C for 30 s, and 68 °C for 30 s. The amplicons were subjected to electrophoresis on 2.0% agarose Tris-borate-EDTA buffer (89 mM Tris-borate, 89 mM boric acid, 2 mM EDTA) gels.

### Quantification of Z chromosome-linked genes

The effects on dosage compensation were assessed by measuring differences in gene expression, as described in *Fukui et al., 2015* and *Arai et al., 2023b*. A total of 1.0 μg of the total RNA extracted from *Hm-oscar* or *GFP*-overexpressed mature embryos (108 hpo, approximately 40–50 sex-determined embryos) was used to prepare mRNA-seq libraries via the NEBNext Poly (A) mRNA Magnetic Isolation Module (New England Biolabs) and the NEBNext Ultra II RNA Library Prep kit for Illumina (New England Biolabs) following the manufacturer's protocol. The adaptor sequences and low-quality reads (Qscore <20) were removed from the generated sequence data [150 bp paired-end (PE150)] using Trimmomatic (*Bolger et al., 2014*). The trimmed reads were mapped to the previously assembled transcriptome database for *H. magnanima* (*Arai et al., 2023b*) using Kallisto (*Bray et al., 2016*) to generate the normalized read count data (transcripts per million, TPM). While we did not generate new RNA-seq data of the *w*Hm-t-infected and uninfected groups, the expression data included in *Arai et al., 2023b* were re-analyzed along with the newly obtained RNA-seq data above. The binary logarithms of the TPM differences between males and females belonging to each *H. magnanima* group (i.e. *Hm-oscar*/*GFP*-expressed) were calculated to assess the fold-changes in gene expression levels between sexes. As described in *Arai et al., 2023b*, the transcriptome data of *H. magnanima* were annotated using the *B. mori* gene sets obtained from KAIKOBASE (https://kaikobase.dna.affrc.go.jp). The binary logarithms of the TPM differences between males and females in the *B. mori* chromosomes 1–28 were plotted.

To confirm the effects observed in the RNA-seq-based quantification of the Z chromosome genes, expression level of the Z-linked *Hmtpi* gene was quantified by reverse transcription-qPCR as described in *Arai et al., 2023b*. Briefly, total RNA (100–300 ng) extracted from pooled male or female mature embryos (108 hpo) was reverse transcribed using PrimeScript II reverse transcriptase (TaKaRa) as described above. The cDNA was used to quantify relative gene expression levels, with normalization to the control gene elongation factor 1 a (*ef1a*). qPCR was performed using primer sets for *Hmtpi* (HmTpi_F180702_212: 5′-GCTGCGAGTGGGTGATTTTG-3′; HmTpi_R180702_326: 5′-GCGATCACTTTCAGACCCGA-3′) and *Hmef1a* (Hmef1a_F_val1_85: 5′-TTTCCAGGGTGGTTGAGCA-3′; Hmef1a_R_val1_193: 5′-CCGTTAAGGAGCTGCGTCG-3′), and KOD SYBR qPCR Mix (Toyobo, Osaka, Japan) in a LightCycler 96 system (Roche, Basel, Switzerland). The qPCR reaction was made with 5 μl KOD SYBR (TOYOBO), 0.4 μL forward primer [10 pmol/μl], 0.4 μl reverse primer [10 pmol/μl], and 3.2 μl water. The qPCR was performed under the following conditions: 180 s at 95 °C, 40 cycles of 8 s at 98 °C, 10 s at 60 °C, and 10 s at 68 °C, followed by heating to 90 °C for melting curve analysis.

Mean cycle threshold ($C_T$) values of samples were calculated for at least 13 replicates, and both $\Delta C_T$ ($C_T$ Z gene – $C_T$ ef1a) and $\Delta\Delta C_T$ ($\Delta C_T$male – $\Delta C_T$fem) values were calculated.

## Transfection assays and quantification of *BmIMP^M*

The coding sequence of *Hmmasc* was amplified from cDNA derived from the RNA extracted from male embryos of the NSR line using KOD-one (TOYOBO) with the following primer set: HmMasc_CDS_HindIIIF: 5′- GCAAAGCTTCAACATGATCTCTCGCCAACCACAATCAACATCA-3′ and HmMasc_CDS_BamHI R: 5′- GCAGGATCCCAACCTACTGATAAGGAGGGAAGTAAGGCTGCTG-3′. The following PCR conditions were applied: 20 cycles of 98 °C for 10 s, 62 °C for 5 s, and 68 °C for 15 s. The amplicon was purified with the QIAquick PCR purification kit (Qiagen) and cloned into the pIZ/V5-His vector using the HindIII and BamHI enzymes (New England Biolabs). The codon-optimized *oscar* gene of *w*Fur (*Katsuma et al., 2022*) was also cloned into the pIZ/V5-His vector using the KpnI and NotI enzymes (TaKaRa).

To verify the masculinizing function of *Hmmasc*, BmN-4 cells ($4 \times 10^5$ cells per dish, diameter 35 mm) were transfected with 1 µg of each plasmid DNA (pIZ/V5-His having *Hmmasc*) using FuGENE HD (Promega, WI, USA), as described in *Katsuma et al., 2022*. To clarify whether *Hm-oscar* suppressed the function of *Hmmasc*, 1 µg of plasmid DNA (pIZ/V5-His bearing either *Hmmasc* or *Hm-oscar* and non-inserted vector as negative control) was transfected to the BmN-4 cells using FuGENE HD (Promega). Three days after transfection, the cells were collected and subjected to RNA extraction via TRI REAGENT (Molecular Research Center Inc, USA) and cDNA construction with AMV transcriptase (TaKaRa). The degree of masculinization in the BmN-4 cells (default female-type sex determination) was verified by quantifying the expression levels of *BmIMP^M* (*Suzuki et al., 2010*), which is involved in male-specific sex determination cascades, using primers BmIMP_F: 5′-ATGCGGGAAGAAGGTTTTATG-3′ and BmIMP_R: 5′-TCATCCCGCCTCAGACGATTG-3′, as described in *Fukui et al., 2015*. *BmRp49* gene was also amplified as a control gene using primers rp49_F: 5′-CCCAACATTGGTTACGGTTC-3′ and rp49_R: 5′-GCTCTTTCCACGATCAGCTT-3′. Further interactions between *Hm-oscar* and *masc* genes derived from lepidopteran insects [i.e. *Trilocha varians masc* (*Tvmasc*), *Spodoptera frugiperda masc* (*Sfmasc*), *B. mori masc* (*Bmmasc*), *O. furnacalis masc* (*Ofmasc*), and *Papilio machaon* masc (*Pmmasc*)] (*Katsuma et al., 2022*) were assessed using the same procedures. The expression levels of *BmIMP^M* were analyzed as described in *Katsuma et al., 2022*. In brief, the relative expression of *BmIMP^M* ($C_T$ *BmIMP^M* / $C_T$ *BmRp49*) was calculated for each experimental group. Then, relative expression of *BmIMP^M* of each sample was normalized by setting the mean in the *Hmmasc* singly transfected group as 100. The normalized relative expression of *BmIMP^M* was visualized using R v4.0 with ggplot2 (https://ggplot2.tidyverse.org/).

## Quantification and statistical analysis

The number of surviving *H. magnanima* injected with either *Hm-oscar, wmk*s, or *GFP* in each replicate were counted at the adult stage. The male ratios (number of adult males/numbers of all adults) under all conditions were compared using the Steel–Dwass test in R v4.0. The sex ratio bias was also assessed based on the total numbers of (i) male and female adults and (ii) male and female embryos under each condition using the binomial test in R v4.0.

Gene expression levels between chromosomes (RNA-seq), a Z-linked *Hmtpi* gene expression in *H. magnanima* (RT-qPCR), and normalized *BmIMP^M* expression in BmN-4 cells under all conditions (RT-qPCR) were compared using the Steel–Dwass test in R v4.0.

A phylogenetic tree of Masc proteins used in this study was constructed based on maximum likelihood with bootstrap re-sampling of 1000 replicates using MEGA7 (*Kumar et al., 2016*).

## Acknowledgements

We thank Greg Hurst (Institute of Infection, Veterinary and Ecological Sciences, University of Liverpool, Liverpool, UK), and Takeshi Suzuki (Graduate School of Bio-Applications and Systems Engineering, Tokyo University of Agriculture and Technology, Tokyo, Japan) for their kind advice. We acknowledge support from the Japan Society for the Promotion of Science (JSPS) Research Fellowships for Young Scientists [Grant Number 19J13123 and 21J00895 to H Arai], JSPS Grant-in-Aid for Scientific Research [Grant Number 22K14902 to H Arai, 23H02229 and 24H02293 to D Kageyama, 22H00366 and 24H02289 to S Katsuma], JSPS Fund for the Promotion of Joint International Research

(Fostering Joint International Research (B)) [Grant Number 21KK0105 to H Arai and MN Inoue]. The wHm-t-infected *H. magnanima* was collected from Tea Research and Extension Station (Taoyuan City, Taiwan), and imported with permission from the Ministry of Agriculture, Forestry and Fisheries (No. 27 - Yokohama Shokubou 891 and No. 297 - Yokohama Shokubou 1326). All collaborators are presented as co-authors, and the results have been shared with the provider communities. Moreover, our group is committed to international scientific partnerships as well as institutional capacity building.

## Additional information

### Funding

| Funder | Grant reference number | Author |
|---|---|---|
| Japan Society for the Promotion of Science | 19J13123 | Hiroshi Arai |
| Japan Society for the Promotion of Science | 21J00895 | Hiroshi Arai |
| Japan Society for the Promotion of Science | 22K14902 | Hiroshi Arai |
| Japan Society for the Promotion of Science | 23H02229 | Daisuke Kageyama |
| Japan Society for the Promotion of Science | 24H02293 | Daisuke Kageyama |
| Japan Society for the Promotion of Science | 22H00366 | Susumu Katsuma |
| Japan Society for the Promotion of Science | 24H02289 | Susumu Katsuma |
| Japan Society for the Promotion of Science | 21KK0105 | Hiroshi Arai Maki N Inoue |

The funders had no role in study design, data collection and interpretation, or the decision to submit the work for publication.

### Author contributions

Hiroshi Arai, Conceptualization, Resources, Data curation, Formal analysis, Funding acquisition, Validation, Investigation, Visualization, Methodology, Writing – original draft, Project administration, Writing – review and editing; Susumu Katsuma, Resources, Funding acquisition, Investigation, Writing – review and editing; Noriko Matsuda-Imai, Investigation, Methodology, Writing – review and editing; Shiou-Ruei Lin, Maki N Inoue, Resources, Writing – review and editing; Daisuke Kageyama, Conceptualization, Supervision, Funding acquisition, Project administration, Writing – review and editing

### Author ORCIDs

Hiroshi Arai https://orcid.org/0000-0001-9912-3489
Daisuke Kageyama https://orcid.org/0000-0002-9026-9825

Reviewer #1 (Public review): https://doi.org/10.7554/eLife.101101.4.sa1
Reviewer #2 (Public review): https://doi.org/10.7554/eLife.101101.4.sa2
Author response https://doi.org/10.7554/eLife.101101.4.sa3

## Additional files

### Supplementary files

MDAR checklist

## Data availability

High-throughput sequencing data are available under accession numbers DRA018708 and PRJDB18169 (BioProject). All data generated during this study are included in the manuscript and a supporting Excel file. Further information and requests for resources and reagents are accessible from the lead contact, Hiroshi Arai (dazai39papilio@gmail.com/HArai@liverpool.ac.uk).

The following dataset was generated:

| Author(s) | Year | Dataset title | Dataset URL | Database and Identifier |
|---|---|---|---|---|
| Arai H, Katsuma S, Matsuda-Imai N, Lin SR, Inoue MN, Kageyama D | 2025 | Transcriptome data for *Homona magnanima* | https://www.ncbi.nlm.nih.gov/bioproject/?term=PRJDB18169 | NCBI BioProject, PRJDB18169 |

The following previously published dataset was used:

| Author(s) | Year | Dataset title | Dataset URL | Database and Identifier |
|---|---|---|---|---|
| Arai H, Takamatsu T, Lin SR, Mizutani T, Omatsu T, Katayama Y, Nakai M, Kunimi Y, Inoue MN | 2023 | Distinct effects of the male-killing bacteria, *Wolbachia* and *Spiroplasma*, and a partiti-like virus in the tea pest moth, *Homona magnanima* | https://www.ncbi.nlm.nih.gov/bioproject/?term=PRJDB13118 | NCBI BioProject, PRJDB13118 |

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
