## [Editor Report · eLife Assessment]

Hardly anything is known about the genetic basis and mechanism of male-killing. Recently, a gene called oscar, in the bacterium *Wolbachia*, was implicated in killing male corn borer moths by interfering with moth genes that control sex determination and proper dosage of sex-specific genes. In this paper, the authors show that a distantly related oscar gene in another strain of *Wolbachia* kills male tea tortrix moths in a similar mechanism. This **valuable** study cements our understanding of the sophisticated way that *Wolbachia* kills male moths and butterflies (Lepidoptera) so early in their development. The conclusions are supported by **solid** evidence.

---

## [Referee Report · Reviewer #1 (Public review)]

Summary:

Insects and their relatives are commonly infected with microbes that are transmitted from mothers to their offspring. A number of these microbes have independently evolved the ability to kill the sons of infected females very early in their development; this male killing strategy has evolved because males are transmission dead-ends for the microbe. A major question in the field has been to identify the genes that cause male killing and to understand how they work. This has been especially challenging because most male-killing microbes cannot be genetically manipulated. This study focuses on a male-killing bacterium called *Wolbachia*. Different *Wolbachia* strains kill male embryos in beetles, flies, moths, and other arthropods. This is remarkable because how sex is determined differs widely in these hosts. Two *Wolbachia* genes have been previously implicated in male-killing by *Wolbachia*: oscar (in moth male-killing) and wmk (in fly male-killing). The genomes of some male-killing *Wolbachia* contain both of these genes, so it is a challenge to disentangle the two.

This paper provides strong evidence that oscar is responsible for male-killing in moths. Here, the authors study a strain of *Wolbachia* that kills males in a pest of tea, Homona magnanima. Overexpressing oscar, but not wmk, kills male moth embryos. This is because oscar interferes with masculinizer, the master gene that controls sex determination in moths and butterflies. Interfering with the masculinizer gene in this way leads the (male) embryo down a path of female development, which causes problems in regulating the expression of genes that are found on the sex chromosomes.

Strengths:

The authors use a broad number of approaches to implicate oscar, and to dissect its mechanism of male lethality. These approaches include: (a) overexpressing oscar (and wmk) by injecting RNA into moth eggs, (b) determining the sex of embryos by staining female sex chromosomes, (c) determining the consequences of oscar expression by assaying sex-specific splice variants of doublesex, a key sex determination gene, and by quantifying gene expression and dosage of sex chromosomes, using RNASeq, and (d) expressing oscar along with masculinizer from various moth and butterfly species, in a silkmoth cell line. This extends recently published studies implicating oscar in male-killing by *Wolbachia* in Ostrinia corn borer moths, although the Homona and Ostrinia oscar proteins are quite divergent. Combined with other studies, there is now broad support for oscar as the male-killing gene in moths and butterflies (i.e. order Lepidoptera).

---

## [Referee Report · Reviewer #2 (Public review)]

*Wolbachia* are maternally transmitted bacteria that can manipulate host reproduction in various ways. Some *Wolbachia* induce male killing (MK), where the sons of infected mothers are killed during development. Several MK-associated genes have been identified in Homona magnanima, including Hm-oscar and wmk-1-4, but the mechanistic links between these *Wolbachia* genes and MK in the native host are still unclear.

In this manuscript, Arai et al. show that Hm-oscar is the gene responsible for *Wolbachia*-induced MK in Homona magnanima. They provide evidence that Hm-Oscar functions through interactions with the sex determination system. They also found that Hm-Oscar disrupts sex determination in male embryos by inducing female-type dsx splicing and impairing dosage compensation. Additionally, Hm-Oscar suppresses the function of Masc. The manuscript is well-written and presents intriguing findings. The results support their conclusions regarding the diversity and commonality of MK mechanisms, contributing to our understanding of the mechanisms and evolutionary aspects of *Wolbachia*-induced MK.

---

## [Author Response]

The following is the authors’ response to the previous reviews.

**Public Reviews:**

**Reviewer #1 (Public review):**
Summary:Insects and their relatives are commonly infected with microbes that are transmitted from mothers to their offspring. A number of these microbes have independently evolved the ability to kill the sons of infected females very early in their development; this male killing strategy has evolved because males are transmission dead-ends for the microbe. A major question in the field has been to identify the genes that cause male killing and to understand how they work. This has been especially challenging because most male-killing microbes cannot be genetically manipulated. This study focuses on a male-killing bacterium called *Wolbachia*. Different *Wolbachia* strains kill male embryos in beetles, flies, moths, and other arthropods. This is remarkable because how sex is determined differs widely in these hosts. Two *Wolbachia* genes have been previously implicated in male-killing by *Wolbachia*: oscar (in moth male-killing) and wmk (in fly male-killing). The genomes of some male-killing *Wolbachia* contain both of these genes, so it is a challenge to disentangle the two.This paper provides strong evidence that oscar is responsible for male-killing in moths. Here, the authors study a strain of *Wolbachia* that kills males in a pest of tea, Homona magnanima. Overexpressing oscar, but not wmk, kills male moth embryos. This is because oscar interferes with masculinizer, the master gene that controls sex determination in moths and butterflies. Interfering with the masculinizer gene in this way leads the (male) embryo down a path of female development, which causes problems in regulating the expression of genes that are found on the sex chromosomes.

We would like to thank you for evaluating our manuscript.

Strengths:The authors use a broad number of approaches to implicate oscar, and to dissect its mechanism of male lethality. These approaches include: (a) overexpressing oscar (and wmk) by injecting RNA into moth eggs, (b) determining the sex of embryos by staining female sex chromosomes, (c) determining the consequences of oscar expression by assaying sex-specific splice variants of doublesex, a key sex determination gene, and by quantifying gene expression and dosage of sex chromosomes, using RNASeq, and (d) expressing oscar along with masculinizer from various moth and butterfly species, in a silkmoth cell line. This extends recently published studies implicating oscar in male-killing by *Wolbachia* in Ostrinia corn borer moths, although the Homona and Ostrinia oscar proteins are quite divergent. Combined with other studies, there is now broad support for oscar as the male-killing gene in moths and butterflies (i.e. order Lepidoptera). So an outstanding question is to understand the role of wmk. Is it the master male-killing gene in insects other than Lepidoptera and if so, how does it operate?

We would like to thank you for evaluating our manuscript. Our data demonstrated that Oscar homologs play important roles in male-killing phenotypes in moths and butterflies; however, the functional relevance of wmk remains uncertain. As you noted, whether wmk acts as a male-killing gene in insects such as flies and beetles—or even in certain lepidopteran species—requires further investigation using diverse insect models, which we are eager to explore in future research.

Weaknesses:I found the transfection assays of oscar and masculinizer in the silkworm cell line (Figure 4) to be difficult to follow. There are also places in the text where more explanation would be helpful for non-experts.

Thank you for your suggestion. We have revised the section on the cell-based experiment. Further, we revised the manuscript to make it accessible to a broader audience. We believe these revisions have significantly improved the clarity and comprehensiveness of our manuscript.

**Reviewer #2 (Public review):**
Summary:*Wolbachia* are maternally transmitted bacteria that can manipulate host reproduction in various ways. Some *Wolbachia* induce male killing (MK), where the sons of infected mothers are killed during development. Several MK-associated genes have been identified in Homona magnanima, including Hm-oscar and wmk-1-4, but the mechanistic links between these *Wolbachia* genes and MK in the native host are still unclear.In this manuscript, Arai et al. show that Hm-oscar is the gene responsible for *Wolbachia*-induced MK in Homona magnanima. They provide evidence that Hm-Oscar functions through interactions with the sex determination system. They also found that Hm-Oscar disrupts sex determination in male embryos by inducing female-type dsx splicing and impairing dosage compensation. Additionally, Hm-Oscar suppresses the function of Masc. The manuscript is well-written and presents intriguing findings. The results support their conclusions regarding the diversity and commonality of MK mechanisms, contributing to our understanding of the mechanisms and evolutionary aspects of *Wolbachia*-induced MK.

We would like to thank you for evaluating our manuscript.

Comments on revisions:The authors have already addressed the reviewer's concerns.

We would like to thank you for evaluating our manuscript.

**Reviewer #3 (Public review):**
Summary:Overall, this is a clearly written manuscript with nice hypothesis testing in a non-model organism that addresses the mechanism of *Wolbachia*-mediated male killing. The authors aim to determine how five previously identified male-killing genes (encoded in the prophage region of the wHm *Wolbachia* strain) impact the native host, Homona magnanima moths. This work builds on the authors' previous studies in which(1) they tested the impact of these same wHm genes via heterologous expression in *Drosophila melanogaster*(2) also examined the activity of other male-killing genes (e.g., from the wFur *Wolbachia* strain in its native host: Ostrinia furnacalis moths).Advances here include identifying which wHm gene most strongly recapitulates the male-killing phenotype in the native host (rather than in *Drosophila*), and the finding that the Hm-Oscar protein has the potential for male-killing in a diverse set of lepidopterans, as inferred by the cell-culture assays.

We would like to thank you for evaluating our manuscript.

Strengths:Strengths of the manuscript include the reverse genetics approaches to dissect the impact of specific male-killing loci, and use of a "masculinization" assay in Lepidopteran cell lines to determine the impact of interactions between specific masc and oscar homologs.

We would like to thank you for evaluating our manuscript.

Weaknesses:It is clear from Figure 1 that the combinations of wmk homologs do not cause male killing on their own here. While I largely agree with the author's conclusions that oscar is the primary MK factor in this system, I don't think we can yet rule out that wmk(s) may work synergistically or interactively with oscar in vivo. This might be worth a small note in the discussion. (eg at line 294 'indicating that wmk likely targets factors other than masc." - this could be downstream of the impacts of oscar; perhaps dependent on oscar-mediated impacts on masc first).

We sincerely appreciate your suggestion. Whilst wmk genes themselves did not exhibit apparent lethal effects on the native host, as you noted, we cannot entirely rule out the possibility that wmk may be involved in male-killing actions, either directly or indirectly assisting the function of Hb-oscar. Following your suggestion, we have added a brief note in the discussion section regarding the interpretation of wmk functions.

“In addition, Katsuma et al. (2022) reported that the wmk homologs encoded by wFur did not affect the masculinizing function of masc in vitro, indicating that wmk likely targets factors other than masc. Whilst we cannot rule out the possibility that wmk may work synergistically or interactively with oscar in vivo—potentially acting downstream of oscar’s impact—our results strongly suggested that *Wolbachia* strains have acquired multiple MK genes through evolution.” (lines 287-292)

Regarding the perceived male-bias in Figure 2a: I think readers might be interpreting "unhatched" as "total before hatching". You could eliminate ambiguity by perhaps splitting the bars into male and female, and then within a bar, coloring by hatched versus unhatched. But this is a minor point, and I think the updated text helps clarify this.

Thank you for your suggestion. We have accordingly revised the figure 2a. In addition, we have included more detailed information in the first sentence of the section Males are killed mainly at the embryonic stage.

“The sex of hatched larvae (neonates) and the remaining unhatched embryos was determined by the presence or absence of W chromatin, a condensed structure of the female-specific W chromosome observed during interphase.” (lines 171-173)

The new Figure 4b looks to be largely redundant with the oscar information in Figure 1a.

Thank you for your suggestion. We have removed Figure 4b due to its overlap with Figure 1a and have incorporated relevant figure legends into the Figure 1a legend.

Updated statistical comparisons for the RNA-seq analysis are helpful. However these analyses are based on single libraries (albeit each a pool of many individuals), so this is still a weaker aspect of the manuscript.

Thank you for your suggestion. As you noted, the use of single libraries (due to the limited number of available individuals, though each includes approximately 50 males and females) may be a potential limitation of this study. However, as demonstrated in the qPCR assay for the Z-linked gene provided in the previous revision, we believe that our data and conclusion—that *Wolbachia*/ Hb-oscar disrupts dosage compensation by causing the overexpression of Z-linked genes—are well-supported and robust.

The new information on masc similarity is useful (Fig 4d) - if the authors could please include a heatmap legend for the colors, that would be helpful. Also, please avoid green and red in the same figure when key for interpretation.

Thank you for your suggestion. We have accordingly included a heatmap legend and revised the colors.

Figure 1A "helix-turn-helix" is misspelled. ("tern").

We have revised.

**Recommendations for the authors:**
Comments from the reviewing editor: I would suggest you address the comments of the reviewer on the revised version.

We have further revised the manuscript to address all the questions, comments and suggestions provided by the reviewers. We believe that the resulting revisions have significantly enhanced the quality and comprehensiveness of our manuscript.

**Reviewer #1 (Recommendations for the authors):**
Thank you for revising this manuscript. I have a few last recommendations:- Line 214: re: 'Statistical data are available in the supplementary data file', it would be more helpful to add a few words here that actually summarize the statistical results

We would like to thank you for your suggestion. We have revised the sentence to describe the overview of the statistical results.

“RNA-seq analysis revealed that, in Hm-oscar-injected embryos, Z-linked genes (homologs on the B. mori chromosomes 1 and 15) were more expressed in males than in females (Fig. 3a), which was not observed in the GFP-injected group (Fig. 3b). Similarly, as previously reported by Arai et al. (2023a), high levels of Z-linked gene expression were also observed in wHm-t-infected males, but not in NSR males (Fig. 3c,d). The high (i.e., doubled) Z-linked gene expression in both Hm-oscar-expressed and wHm-t-infected males was further confirmed by quantification of the Z-linked Hmtpi gene (Fig. 3e). These trends were statistically supported, with all data available in the supplementary data file.” (lines 205-213)

- Figure 1 legend: do you mean 'bridged' instead of 'brigged'?

We have accordingly revise, thank you for the suggestion.